# Natural Infections of Potato Plants Grown from Minitubers with Blackleg-Causing Soft Rot Pectobacteriaceae

**DOI:** 10.3390/microorganisms10122504

**Published:** 2022-12-17

**Authors:** Jan van der Wolf, Marjon Krijger, Odette Mendes, Viola Kurm, Jack Gros

**Affiliations:** 1Biointeractions and Plant Health, Wageningen University & Research, Droevendaalsesteeg 1, 6708 PB Wageningen, The Netherlands; 2Agrico Research, Burchtweg 17, 8314 PP Bant, The Netherlands

**Keywords:** *Pectobacterium*, *Dickeya*, TaqMan, infection source, airborne infection, soilborne infection, initial infection, *gapA* sequencing

## Abstract

Information on the infection incidence of blackleg-causing soft rot Pectobacteriaceae (BL-SRP) in potato crops grown from minitubers (PB1-crop) and the distribution of BL-SRP in individual plants was collected during a two-year survey conducted at five potato growers located in the Netherlands. In the last weeks before haulm destruction, leaves, stems, and tubers of 100 or 200 plants were analyzed separately for the presence of *Pectobacterium parmentieri*, *P. brasiliense*, *P. atrosepticum*, and *Dickeya* spp. Extracted plant parts enriched for BL-SRP were analyzed with TaqMan assays specific for the detection of blackleg-causing BL-SRP. In 2019, low incidences of *P. parmentieri* (1–6%) in leaves were found at four growing sites. At one farm, reactions were detected in TaqMan assays for *D. zeae* and *D. chrysanthemi* in leaves. In 2020, the crops of two growers were largely free from BL-SRP. At one farm, a high infection incidence (21%) was found for *D. fangzhongdai* in tubers. The isolated pathogen was able to cause potato blackleg. At two other farms, high infection incidences in tubers were found with *P. brasiliense* (35–39%) and *P. parmentieri* (12–19%), whereas the incidence of *P. brasiliense* in leaves was also high (8%). In conclusion, high infection incidences with BL-SRP in potatoes can be found in a PB1 crop at the end of the growing season. Infections in individual plants were found either in tubers or in leaves. The potential sources of initial infection are discussed.

## 1. Introduction

Soft rot Pectobacteriaceae (SRP) comprise the genera *Dickeya* and *Pectobacterium*, responsible for diseases in various important crops worldwide, including potato, tomato, maize, cabbage, and ornamental plants [1,2]. In potatoes, they can cause soft rot, blackleg, and slow wilt, which are responsible for high economic losses due to yield loss, downgrading, and rejection of seed lots [3].

Seed potatoes have been recognized as the most important source of disease development [4,5]. Nevertheless, even after many years of research, knowledge of the source of initial infection is still limited. Seed potato cultivation starts with using minitubers or tubers from clonal selection, virtually free of SRP, but even in the first year of tuber multiplication, infections can occur [6].

Infections may start from soilborne inoculum, but several studies indicated that the pathogen only survives for a limited period in soil, even in association with plant debris. However, it is difficult to conclusively exclude soil as a source of infection as detection of low inoculum densities in bulk soils remains challenging. In addition, the role of nematodes or soilborne arthropods, in which SRP may live for longer periods than in soil, cannot be fully excluded [7].

The most likely pathway for initial infections is via airborne inoculum carried by rain, aerosols, insects, machines, furs, feathers, clothes, footwear, animals, or laborers. The inoculum may be deposited on the haulms, after which wounds or natural openings, such as stomata or hydathodes, may serve as a port of entry to establish infection [8]. The inoculum may also migrate via water from haulms into the soil, after which tuber infections can occur [8]. In theory, transmission from infected haulms could also result in root infections and, consecutively, in infections of stolons and tubers, but this pathway has not been studied yet.

The aim of our studies was to extend our knowledge of the sources responsible for and pathways of SRP resulting in the initial infection of a potato crop. For this, we investigated where the first infections of potato plants grown from SRP-free minitubers occurred—in the tubers, the stem base, or the leaves. A survey was conducted for two years at five potato growers in the Netherlands. We assumed that tuber infections in the absence of leaf infections point to the soil as an inoculum source. Leaf infections without tuber infections indicate an important role in forming airborne inoculum. If both leaves and tubers of individual plants are infected, it is likely that the inoculum migrated from leaf to soil.

## 2. Materials and Methods

### 2.1. Bacterial Strains and Growth Conditions

D. solani IPO2222, *P. atrosepticum* IPO1007, *P. parmentieri* IPO1955, and *P. brasiliense* IPO3649 were used to develop the multiplex TaqMan assay and the evaluation of the enrichment procedures. D. solani IPO2222 was used as a positive control during isolation procedures. Isolates were stored at −80 °C on beads (to protect bacterial preservers; http://www.tsc-swabs.co.uk; accessed on 1 December 2022). Unless otherwise stated, bacteria were grown on tryptone soya agar (TSA; Oxoid) for 48–72 h at 27 °C. To isolate bacteria from plant extracts, the plant material was spread-plated in tenfold serial dilutions on a double-layer crystal violet pectate (DL-CVP) medium [9].

### 2.2. Sampling Plant Material

In a two-year survey (2019 and 2020), plant material from a potato grown from minitubers of cultivar Agria, a cultivar susceptible to SRP pathogens, was collected at the end of the growing season from five commercial seed potato growers. Two growers were located in the central part of the Netherlands (Noordoostpolder), and the other three were in the Western part of the Netherlands (Noord-Holland) (Table 1). Minitubers were planted in ridges between 15 and 30 April, with a planting distance of 25 cm between tubers and 40 cm between rows. In 2019, 100 plants and in 2020, 200 plants per grower were sampled and individually analyzed. In most cases, samples were collected in dry weather conditions.

Three samples per plant were collected separately, i.e., the top leaves, the stem base, and the progeny tubers. The stem base was sampled and analyzed to test for systemic infections in plants. All stems from a plant were held in the middle and cut 10 cm above the soil splashing area (to prevent soil contaminations). Pruning shears were cleaned between each cutting and disinfected with 70% ethanol to prevent cross-contamination. Every 2–3 plants in a row were sampled. The sampling was only performed from the edge rows and along spraying paths. The 20 cm of the leaves at the top of the plant were transferred to a labeled plastic bag suitable for vacuum (LDPE bags, transparent, 0.10 mm, 300 × 500 mm, VPP Packaging B.V., Bussum, NL) and, after cleaning the pruning shears, 10 cm of the stem bases were transferred to another labeled vacuum plastic bag. Progeny and mother tubers (if present) were collectively sampled (maximally 10 tubers per plant), but rotten mother tubers were collected separately. Samples were transported at room temperature and stored in a cold room (4 °C) overnight to be processed the next day.

### 2.3. Pathogen Detection Procedure

A protocol was developed based on incubation of samples under vacuum (anoxic) conditions, favoring selective growth of SRP, and pathogen detection by a (multiplex) TaqMan able to detect all blackleg-causing SRP (BL-SRP) present in the Netherlands, i.e., *P. atrosepticum* (Pat), *P. parmentieri* (Ppa), *P. brasiliense* (Pbr), and a generic *Dickeya* sp. assay as outlined below. In case of a positive reaction in the first (multiplex) TaqMan assay, the blackleg-causing species were identified by a second TaqMan assay, specific for the individual species.

### 2.4. Enrichment Procedure

The leaf material, which was in the field directly collected into a vacuum bag, was weighed, and 100 mL PEB (MgSO_4_.7H_2_O, 0.3 g; (NH_4_)_2_SO_4_, 1.0 g; K_2_HPO_4_.3H_2_O, 1.31 g; Polygalacturonic acid, sodium salt (Merck Life Science, Amsterdam, The Netherlands), 1.5 g; in 1 L demineralized water; PH 7.2) were added to each bag. The stem material was weighed, and 50 mL PEB was added to each bag. The tubers were washed with tap water, placed in a vacuum bag, and weighed. Five mL of tap water per tuber was added. Bags were carefully folded, closed with clothespins without damaging them, and placed in a box. A box was then incubated for 30 min at 25 °C while shaking at 130 rpm. The vacuum was applied to the bags, which were then incubated flat at 25 °C in a box in the climate room for 5 days. After enrichment, samples were manually crushed, and a sample of 1 mL extract was taken out of each bag and transferred into a collection microtube of a 8-well strip placed in a 96-well blue microtube rack (19560, Qiagen Benelux B.V., Venlo, The Netherlands). The extracts were centrifuged for 15 min at 6000 rpm, and the supernatant was removed. Tubes were then sealed with 8-well strip caps (19566, Qiagen Benelux B.V., Venlo, The Netherlands), and the pellets were stored at −20 °C. Per each experiment, leaf, stem, and tuber extracts were supplemented with 2.10^3^ cfu/mL of Ppar IPO1955, Patr IPO1007, Pbr IPO3649, or Dsol IPO2222 to test for enrichment.

To support the isolation of SRP from TaqMan-positive samples, in 2019, prior to enrichment, a sample of 100 μL of the leaf, stem, or tuber extract was transferred from the vacuum bag to a 2 mL tube with 900 μL PEB with screw cap and incubated for 1 day at 21 °C. After incubation, 500 μL of 60% (*v*/*v*) glycerol was added to each of the 2 mL tubes and stored at −80 °C. Using this procedure, no target bacteria detected in the TaqMan assays could be isolated. Therefore, in 2020, the procedure was modified. After enrichment of plant tissues for five days, 100 µL of liquid that leaked from rotten plant material was collected, mixed with 50 µL of 60% glycerol, and stored at −80 °C in microtiter plates for an eventual reisolation.

### 2.5. DNA Extraction Procedure

Prior to the DNA extraction, a suspension of Acidovorax cattleya (Acat) was supplemented as extraction and amplification control. In the case of efficient extraction and amplification, for the Acat assay, Ct-values were expected at 30–32 [10]. DNA extraction was performed using the DNA extraction sbeadex maxi plant kit (NAP41620, LGC Genomic GMGH, Berlin, Germany) in combination with the KingFisher™ Flex Purification System (Automata Technologies, London, UK). To each sample, a chrome steel ball of 3.2 mm diameter was added as well as the lysis buffer supplemented with 1 µg RNase A and 50 µL of the internal extraction control. The internal control was prepared by diluting 10 μL of the Acat cells suspension with an optical density of 600 nm of 0.8 in 10 mL PBS 0.01 M. The samples were homogenized using the Tissuelyser II Qiagen twice for 20 s at 20 Hz, the plate being inverted between the 2 bead-beating steps. The DNA extraction protocol was performed according to the manufacturer’s instructions. The samples were eluted in 100 µL.

### 2.6. Multiplex TaqMan Assay for Blackleg-Causing Pathogens

A multiplex TaqMan assay was used to target all important blackleg-causing pathogens present in Europe (*P. atrosepticum*, *P. Brasiliense*, *P. parmentieri*, and *Dickeya* sp.). The probes of all 4 targets were labeled FAM, and the Acat probe of the internal control was labeled HEX (Table 2). The reaction mix consisted of 5 µL of the PerfeCTa Multiplex qPCR ToughMix, Low ROX enzyme mix (95149-250, QuantaBio, Beverly, MA, USA), 0.3 µM *Dickeya* Fw284, 0.3 µM *Dickeya* Rv284, 0.1 µM *Dickeya* P284, 0.3 µM ECA-CSL-1F, 0.3 µM ECA-CSL-89R, 0.1 µM ECA-CSL-36T-P, 0.12 µM PbrFw, 0.3 µM PbrRv, 0.1 µM PbrP, 0.3 µM PwF1, 0.3 µM PwR1, 0.1 µM PwP1, 0.3 µM Acat 2-F, 0.3 µM Acat 2-R, 0.1 µM Acat 2-Pr, water, and 2 µL sample of 25 µL. The amplification was performed in the QuantStudio (AppliedBiosystems, Waltham, MA, USA) with the following program: after a hold of 2 min at 95 °C, the amplification took place during 40 cycles of 15 s at 95 °C and 60 s at 60 °C. Data analysis was done by automatic threshold calculation within the QuantStudio 12KFlexSoftware v1.3. Only reactions with Ct-values below 35 and a typical logarithmic curve were considered positive. The multiplex assay was compared with simplex assays using gBlocks comprising the target sequences (Table 3) as well as using genomic DNA of pure cultures of target bacteria. Positive samples in the multiplex assay were tested with the individual TaqMan assays to identify blackleg-causing pathogens.

### 2.7. Species-Specific Simplex TaqMan Assays

In total, 12 simplex TaqMan assays (Table 2) were used to identify SRP positive in the multiplex TaqMan samples or bacteria cultured on plates. The reaction mix of 25 µL consisted of 5 µL of the PerfeCTa Multiplex qPCR ToughMix, Low ROX enzyme mix (95149-250, QuantaBio), 0.3 µM primers, 0.1 µM probe, water, and 2 µL sample (gBlocks or purified DNA from plant material). Amplification and data analysis was done as described for the multiplex TaqMan assays.

### 2.8. Triplex TaqMan Assay for a Clade of Virulent P. brasiliense Strains

P. brasiliense is a heterogeneous species comprising a genetically homogeneous clade in which most strains can cause blackleg, while most strains outside this clade cannot. To specifically detect the group of virulent (vPbr) strains, a triplex TaqMan assay was used based on two loci (LZI and TIR) (Table 2). The TaqMan assay for *P. brasiliense* described by [11] was used to detect all *P. brasiliense* strains. The reaction mix with 25 µL consisted of 5 µL of the PerfeCTa Multiplex qPCR ToughMix, Low ROX enzyme mix (95149-250, QuantaBio), 0.3 µM of each primer, 0.1 µM of each probe, water, and 2 µL sample. Amplification and data analysis was done as described for the multiplex TaqMan assays.

**Table 2 microorganisms-10-02504-t002:** Primers and probes used in this study.

Assay	Strain	Name	Sequence	Target	Reporter Dye	Addition	Reference
* and **	*Dickeya* sp.	*Dickeya* Fw284	tgtgcgttttcgggctactc	potassium transporter Kup			[12]
		*Dickeya* Rv284	ccctgtcttctgttatcaattcattaac				
		*Dickeya* P284	aaccagaataaggccc		FAM	MGB-NFQ	
* and **	*Pectobacterium atrosepticum*	ECA-CSL-1F	cggcatcataaaaacacgcc	unknown			[13]
		ECA-CSL-89R	cctgtgtaatatccgaaaggtgg				
		ECA-CSL-36T-P	acattcaggctgatattccccctgc		FAM	ZEN/IBFQ	
* and **	*P. brasiliense*	PcbrFw	tgcgggttctgcgtttc	*araC*			[14]
		PcbrRv	tggcgcgttcgcaatat				
		PcbrP	caaggcacgatacg		FAM	MGB-NFQ	
* and **	*P. parmentieri*	PwF1	tctgttcaatgtcaacgcaggta	*mdh*			[14]
		PwR1	aggtaaccgcaatttgctcaa				
		PwP1	tgtgcgcaacctg		FAM	MGB-NFQ	
*	*Acidovorax cattleya*	Acat 2-F	tgtagcgatccttcacaag	Unknown			[10]
		Acat 2-R	tgtcgatagatgctcacaat				
		Acat 2-Pr	cttgctctgcttctctatcacg		HEX		
**	*D. dianthicola*	DdiFw	gccgtatccatcatgcttacc	*dnaX*			[15]
		DdiRv	aacgggcgatagtcgtcttg				
		DdiP	tttccggcactcgg		FAM	MGB-NFQ	
**	*D. chrysanthemi*	Fw2	cgatttcccggcaagtgt	*dnaX*			[15]
		Rv2	tggcaaaagggctgaattg				
		LNA probe 3	cgccgTCActccc		FAM	LNA	
**	*D. zeae*	Fw4	tcccgcactaaagttgaaga	*dnaX*			[15]
		R43	gcgagctggcgcgtatt				
		probe	cgcgagactTACtggataacgt		FAM	LNA	
**	*D. solani*	SOL-C-F	gcctacaccatcagggctat	Unknown			[16]
		SOL-C-R	acactacagcgcgcataaac				
		SOL-C-P	ccaggccgtgctcgaaatcc		FAM		
**	*D. dadantii*	Fw2	cccggtttcgcaattcag	*dnaX*			[15]
		Rv3	gggcgtaggcaagacgacta				
		Probe	tttcgccAACaaacggg		FAM	LNA	
**	*D. dieffenbachiae*	Fw2	gaattgcgaaaccgggatta	*dnaX*			[15]
		Rv1	gatttcccggcaggtatcg				
		Probe	cggctaCACcctgc		FAM	LNA	
**	*D. fangzhongdai*	DfF	cttcgccgcccaggtatttt	*fusA*			[17]
		DfR	atcagggcgtgaccttcgtt				
		DfP	tgctgcagactcgatcaggttctga		FAM	ZEN/IBFQ	
***	*P. brasiliense* (virulent group)	LZI_F1	cggtaagttatgccgcatct				unpublished data
		LZI_R1	cactgatctctttcatttagccatatc				(Van der Lee, Van Gent, WUR)
		LZI_P1	tggcattacagaattcattgccaac	Lysozyme inhibitor	FAM	ZEN/IBFQ	
***	*P. brasiliense* (virulent group)	TIR-F2	agataaacaagcgagggttga				unpublished data
		TIR-R2	atctatctcccatttcacccaag				(Van der Lee, Van Gent, WUR)
		TIR-P2	aaatacagcctccattagagtttccc	Toll-like receptor	Yakima Yellow	ZEN/IBFQ	
***	*P. brasiliense (in triplex assay)*	Pb1F	ccttaccaagaagatgtgtgttgc				[11]
		Pb2R	cataaacccggcacgct				
		PbPr	caagcgcacctgttgatgtcatgagtg	16–23 s intergenic spacer	Cy5	TAO/IBRQ	

LNA bases are shown in capital letters; * in Multiplex assay; ** in Simplex assay; *** in Triplex assay.

### 2.9. Pathogen Isolation

Samples found positive were plated in tenfold serial dilutions (undiluted to 10^6^ times diluted in Ringers solution; 2.25 g/L NaCl, 0.105 g/L KCl, 0.12 g/L CaCl, and 0.05 g/L Na_2_CO_3_) on DL-CVP plates [9]. *D. solani* IPO222 was plated on DL-CVP as a positive control to check for the formation of cavities in the plates. Plates were incubated for 4 days at 28 °C. Cavity-forming colonies typical for SRP were streaked to pure cultures on TSA.

#### Confirmation of the Identity of Isolates

Isolates from cavity-forming colonies were identified with individual TaqMan assays (Table 2). The identity of *D. fangzhondai* was further confirmed by analysis of *gap*A sequences [18]. An alignment was made in CLC Genomics Workbench 20.04 (Qiagen, Aarhus, Denmark) and imported into MEGA 7.0 software for constructing a phylogenetic tree. A phylogenetic tree was inferred using the Maximum Likelihood method based on the General Time Reversible model with 500 bootstrapping replications. *Gap*A sequences of *D. fangzhongdai*, *D. dadantii*, *D. solani*, *D. dianthicola*, *D. undicola, D. aquatica*, *D. poaceiphila*, *D. chrysanthemi*, *D. zeae*, *D. paradisiaca* present in Genbank were used to construct a phylogenetic tree with sequences of *P. brasiliense* as an out-group.

### 2.10. Tuber Maceration Assay

The ability of the bacterial strains to cause soft rot was tested in a tuber maceration assay as described by Czajkowski et al. (2010) [19], using *D. solani* as the positive control and water as a negative control.

**Table 3 microorganisms-10-02504-t003:** Gblocks were used in this study.

Target	Sequence	Target Gene	bp	Comments
Generic *Dickeya*	GACCACTTTGCCGTTTTCCACCAACAGGTTAATTTTGCAGCCTGACCCACAGTAAGGGCAGACGGTAATTACTTTCTGCATGACATTGCTCTCCTTCATCGTACCGACGGCACGCTCAGAATTTCATTTCCGCGGCTTCATC**G**AGTGCTGCGCGCCGCTGTTTACGTTGGATCATTTCCTGAATGTCCTCCCGGCTGATCAG	formate dehydrogenase	220	
*Pectobacterium parmentieri*	ACCGGGTGTTGCTGTCGATCTGAGCCATATTCCTACAGCAGTGAAGATCAAAGGCTATAGCGGTGAAGACGCTAAACCAGCGCTTGCTGGTGCGGATATTGTGCTGATTTCCGCTGGCGTGGCACGTAAACCTGGTATGGATCGTTCCGATCTGTTCAATGTCAACGCAGGTATTGTGCGCAACCTGGTTGAGCAAATTGCGGTTACCTGCCCGAAAGCCTGCATCGGGATCATTACTAATCCCGTGAACACGACCGTCGCTATTGCAGCCGAAGTGCTG	malate dehydrogenase	280	
*P. atrosepticum*	AGGATTCAGTTAATAATGCAATGGAATAGCAATGTAATAT**CG**AAATCATTGAACGCTTTTATAGAATAGAGAAGGATCGGCATCATAAAAAC**A**CGCCATTAATAAACACATCAACATTCAGGCTGATATTCCCCCTGCCTATTCCACCTTTCGGATATTACACAGGGTACTTCCCTTATTGCCTTCTATTAAATCAG	unknown	210	A = deletion in primer, CG = different from target sequence (AA). Adjusted to increase GC content.
*P. brasiliense*	CGTCAGGTGACCGGTGCCGGATGGGCGAGGTGAATCGTTCTGATTCGCGCCGAATACTGCCAGGCAACATCGGTGAGTGTTGGTCGGATTTCAGCCGTATTGATGAGGATCTGTCGCTGGCGCGTTCGCAATATCGCCCGTATCGTGCCTTGGTGGAAGAAACGCAGAACCCGCATTCGCAGCCGATGCTGATCATGACATTTGCACTG	unknown	215	

### 2.11. Field Bioassay

The ability of bacterial isolates to cause blackleg in a field bioassay was tested as described by De Haan et al. (2008) [20] with few modifications. Seed tubers of cultivar Kondor or cultivar Agria were inoculated on 21 April 2021 with 10^6^ cfu/mL of the pathogens suspended in water, or tubers were mock-inoculated with water. Tubers were vacuum-infiltrated using a Henkovac vacuum machine (Den Bosch, NL) for 30 min, followed by 2 days of drying in a ventilated open space. A virulent strain of P. brasiliense (IPO3649) was used as a positive control. Per treatment, two blocks of 16 plants were planted on 30 April 2021 in a randomized order in an experimental field with sandy soil in Marknesse (NL). After emergence, plants were observed for symptoms biweekly for 2 months. Bacteria were isolated from symptomatic stems using dilution plating on DL-CVP. After growing strains to pure cultures on TSA, the identity was verified by TaqMan assays.

### 2.12. Statistics

The incidence (*I*) of infected tubers in composite samples was estimated using the statistical equation I = {1 − [(N − *p*)/N]1/n} × 100, in which N is the total number of subsamples tested, p is the number of subsamples tested positive, and n is the number of individuals per subsample [21]. For the field bioassay, an analysis of variance was performed on the number of diseased plants using Genstat (VSN International, 2015. Genstat for Windows 18th Edition. VSN International, Hemel Hempstead, UK. Web page: Genstat.co.uk, accessed on 29 June 2022). Fisher’s Least Significant Difference was used as a post hoc test (*p* = 0.05).

## 3. Results

### 3.1. Development of a Generic Multiplex TaqMan Assay for Blackleg-Causing Bacteria

The reaction kinetics of TaqMan assays, among other factors such as target sequences and cycling conditions, also depends on the fluorophore used to label the probe [22]. In pilot experiments using gBlocks with target sequences, the fluorophore FAM exhibited higher delta Rn values (Rn value experimental reaction minus Rn value of the baseline signal), resulting in higher sensitivity than the fluorophores VIC and TexasRed (Appendix A). As we assumed that infection incidences with BL-SRP in a potato crop grown from minitubers were low, we decided first to analyze samples with a multiplex TaqMan assay, in which the different probes for blackleg-causing bacteria present in Europe were all labeled with FAM, to maximize the sensitivity of the assay. The probe for the Acat TaqMan assay was labeled with HEX. A positive reaction in the multiplex assay was followed by an analysis with simplex assays to determine which species were present. Using gBlocks comprising the target sequences (Table 3), the amplification plots showed that the reaction kinetics of the four assays with FAM-labeled probes in a multiplex format were very similar (Appendix A, Appendix A). The detection thresholds of the assays conducted in a simplex and multiplex format for *P. atrosepticum*, *P. Brasiliense*, and *P. parmentieri* using gBlocks was 100 copies per reaction (Appendix A). For *Dickeya* sp., the detection threshold of the simplex assay was 100 copies, and for a multiplex format, 1000 copies. If genomic DNA was used, the multiplex and simplex assays were also very similar, ranging between 10 and 100 fg (Appendix A).

### 3.2. Survey 2019

In the multiplex assay, at growers A, B, C, and E, positive reactions were mainly found in leaves with infection incidences between 3 and 7%, occasionally in stems but never in tubers (Figure 1). Analyses with the simplex assays revealed that infections were predominantly caused by *P. parmentieri*. Ct-values of positive samples in the simplex assay were between 30 and 35 (Appendix A, Appendix A). Incidentally, models positive in the multiplex assay were negative when tested in the simplex assays. At grower D, leaves were infected with *Dickeya* species and the stems with *P. parmentieri*. Relatively low Ct-values were found in the leaf samples with the genus-specific *Dickeya* TaqMan (Appendix A). Results indicated that three leaf samples were infected with *D. chrysanthemi* and two with *D. zeae* (Table 4). All models positive with the genus-specific *Dickeya* TaqMan assay were dilution-plated on DL-CVP but attempts to isolate SRP from samples positive with the genus-specific *Dickeya* TaqMan assay by dilution-plating on DL-CVP were not successful.

Tubers from the minituber potato crops at growers A–D were harvested by the farmer, transported to Wageningen, and tested for blackleg in May 2020, using the same approach as for the field samples. Per grower, in total, 400 tubers were tested in 40 composite samples of 10 tubers. All composite samples of growers A–C were negative, whereas seven out of 40 subsamples of grower D were positive for *P. brasiliense*, indicating a tuber infection incidence of 1.9%. No blackleg-diseased plants were found in any of the five PB1 crops surveyed.

### 3.3. Survey 2020

In 2020, low infection incidences of BL-SRP were found in plant material from growers A and B, not exceeding 1.5% in 200 samples analyzed per grower (Figure 1). At Grower C, a high incidence (22%) was found with *Dickeya* sp., but only in the tuber samples. In leaves and stems, *P. brasiliense* and *P. parmentieri* were found occasionally. At growers D and E, 37% and 39% of the tuber samples harvested were positive for *P. brasiliense*. At these two growers, individual plants yielded, on average, 7.4 and 6.2 tubers, respectively. In addition, at growers D and E, tuber samples were also highly infected with *P. parmentieri* (13 and 20%, respectively). For tubers, the estimated infection incidence was approximately 6%. Relatively high incidences of *P. brasiliense* were also found in leaves at growers D and E (7 and 8%, respectively). Interestingly, plants with leaf infections of *P. brasiliense* or *P. parmentieri* were not infected in tubers and vice versa. In 20% of all SRP-infected samples, co-infections with *P. brasiliense* and *P. parmentieri* occurred, predominantly in tubers at growers D and E. Ct-values of positive samples were low, in particular for *P. brasiliense*, indicating the presence of high bacterial densities after enrichment (Appendix A). Only in the harvest of grower E, 14 of the 200 plants yielded a rotten mother tuber, from which two were infected with *P. brasiliense*. No blackleg-diseased plants were found in any of the five PB1 crops surveyed.

### 3.4. Characterization of the P. brasiliense Strains

In 2020, both *P. brasiliense* and *Dickeya* sp., but not *P. parmentieri*, were successfully isolated from samples collected from the five-day enriched sample extracts, which were stored with glycerol for two months at −20 °C. Samples from growers D and E, positive for *P. brasiliense*, were further tested using the (vPbr) TaqMan assay against the virulent *P. brasiliense* clade [23]. The percentage of samples positive for the vPbr assay was high and ranged between 66 and 100% of the total number of samples infected with *P. brasiliense* (Figure 2).

The presence of the various clades of *P. brasiliense* was confirmed by dilution plating. At four growers, *P. brasiliense* was isolated from tuber, stem, and leaf samples positive in the vPbr TaqMan assay, and at two growers, isolates were derived from leaf and tuber samples negative in the vPbr TaqMan assay (Appendix A). In a field bioassay with vacuum-infiltrated tubers of cv. Kondor, strains IPO4217, IPO4218, IPO4221, and IPO4227, all designated as virulent *P. brasiliense* strains on the basis of the vPbr TaqMan assay results, caused a high disease incidence of 62–82%, comparable with the positive control for vPbr, IPO3649 (82%) (Figure 3). Inoculation with the other *P. brasiliense* strains, negative in the vPbr TaqMan assay, resulted in a disease incidence of 0 to 6%, not significantly different from the water control. In cv. Agria, only with IPO3649, the positive control, the disease prevalence was higher than the water-treated control, which showed a background of blackleg-diseased plants. In general, the disease prevalence in cv. Agria was much lower than in cv. Kondor.

### 3.5. Identification of Dickeya Species

The *Dickeya* sp. strains, isolated from the tubers of grower C, strongly reacted with the *Dickeya* genus-specific TaqMan assay but not with TaqMan assays developed for detecting *D. solani*, *D. dianthicola*, *D. chrysanthemi*, *D. dadantii* and *D. zeae*, respectively. A phylogenetic analysis with *gapA* sequences of two isolates indicated that the tubers were infected with *D. fangzhongdai*, as they clustered with *D. fangzhongdai* strains from orchid, onion, pear, and water streams (Figure 4). A TaqMan assay for detecting *D. fangzhongdai* based on *fusA* sequences (Tian et al., 2020) was positive and further confirmed the identity of the strains. The *D. fangzhongdai* strains from potatoes were able to macerate potato tuber tissue (Appendix A). The blackleg-causing capacity of the two *D. fangzhongdai* strains in a field bioassay with cv. Kondor was comparable to the virulent *P. brasiliense* strains (Figure 3). Vacuum inoculations of tubers of cv. Kondor resulted in a higher disease prevalence than inoculations of cv. Agria, both with *P. brasiliense* and *D. fangzhongdai.*

The *gapA* sequences of two *D. fangzhongdai* strains numbered IPO4215 and IPO4216, were deposited in Genbank (accession numbers OM809171 and OM809172).

## 4. Discussion

A two-year field survey, in which plants grown from minitubers were analyzed for infections with SRP, showed that infection prevalence was strongly dependent on location and year. In 2019, there was a low prevalence of mainly leaf infections with *P. parmentieri*. Surprisingly, *P. brasiliense*, the dominant blackleg-causing agent in the Netherlands in the last five years, was not detected [3]. The TaqMan reactions in the *P. parmentieri* assay were weak, even though the samples were incubated under conditions favoring the growth of SRP, indicating that mainly dead cells were detected. We speculate that although haulms came in contact with airborne inoculum, conditions were unfavorable to establish an infection. As an exception, at grower D, 5% of the leaves were infected with *Dickeya* sp. Strong TaqMan reactions were found, indicating a successful infection. Interestingly, analysis of samples with TaqMan assays for individual *Dickeya* species indicated that the leaves were infected with *D. chrysanthemi* and *D. zeae*. So far, in Europe, blackleg was predominantly caused by *D. dianthicola* and *D. solani* [3]. *D. chrysanthemi* has been associated with potatoes in Taiwan [24] and in the USA, where it caused stem rot [25]. Recently, strains isolated from potatoes in Switzerland and the Netherlands were described, but no data were provided on their ability to cause blackleg or stem rot [26]. *D. zeae* was only described in association with potatoes in Australia, where it caused soft rot but not blackleg [24,27,28]. Tubers and stems were not infected with the *Dickeya* species found in leaves, indicating that infections were airborne. In the Netherlands, *D. chrysanthemi* was isolated from begonia, chrysanthemum, dahlia and Pastinaca sp. [3]. It may be that the pathogen was transmitted from these (ornamental) host plants. However, these crops were not grown in rotation with potatoes in the preceding two years (personal communication grower).

In our studies, minituber seed lots were not tested for BL-SRP prior to planting, but we do not expect them to be the source of infection. Between 2005 and 2008, 25–30 different seed lots of potato minitubers were tested in the Netherlands annually for the presence of BL-SRP, but they were never detected [29]. Moreover, in 2020, growers B, C, D, and E all received minitubers from the same supplier. The PB1 at grower B remained largely free of BL-SRP; at grower C, it became infected with *D. fangzhongdai*, while at growers D and E, infections with *P. brasiliense* and *P. parmentieri* were found in the PB1 crop.

Although BL-SRP was not detected in tubers of the sampled crop at the end of the growing season, at grower D, a high infection incidence with *P. brasiliense* was found after storage. The infections may be a result of low incidences of infected, rotten tubers present in the sampled crop at harvest but may also originate from other crops if machines and equipment used during harvesting and storage were not thoroughly washed and disinfected prior to use [6,30].

In 2020, the weather conditions during the growing period (June and July) for blackleg development were more favorable than in 2019, with a high level of precipitation. At growers A and B, low infection incidences were found, similar to 2019. However, at growers D and E, high incidences of *P. brasiliense* (35–38%) were found, particularly in tubers. More than 10% of the tuber samples were also infected with *P. parmentieri*. In addition, at growers D and E, relatively high incidences of *P. brasiliense* (8%) were found in leaves. *P. brasiliense* is a genetically and phenotypically heterogeneous group in which strains vary in their ability to cause blackleg [23,31]. A high percentage (66–100%) of the samples infected with *P. brasiliense* reacted with a TaqMan assay specific for a genetically homogeneous clade of strains, from which most strains can cause blackleg [23]. *P. parmentieri* strains tested previously in a field bioassay, erroneously described as a virulent group of *P. carotovorum*, were all able to cause blackleg, although the aggressiveness seems to be lower than of other blackleg-causing species and varies between strains [6,20,32,33].

Plants with infected leaves did not have infected tubers and vice versa (results not shown). It is highly unlikely that the minitubers (seed potatoes) were the source of inoculum, as the seed used by growers B, C, D, and E were from the same producer, whereas only at growers D and E high infection rates were found. In addition, the infection incidence of rotten mother tubers was relatively low, 1% compared with an estimated 6% for the progeny tubers. If the minitubers had acted as an infection source, a high incidence of rotten mother tubers would have been expected.

We hypothesize that the presence of nearby located SRP-infected potato crops of a lower class was predominantly responsible for the initial infections. As in most cases, tuber and leaf infections were not found on the same plant, we conclude that transmission pathways resulting in leaf or tuber infections were different. Leaf infections may be due to transmission of the pathogen via insects or splashing rainwater. It has been shown in glasshouse experiments that after infection, SRP populations can build up in leaves to densities of up to 10^6^ cfu per gram of leaf material [8]. Tuber infections may be caused by transmission of the pathogen from (rotten) mother tubers of blackleg-diseased plants via horizontal transport of water in soil. This dissemination pathway has been evidenced before [6]. *D. solani* was transmitted during the growing season from an infected plant to the third plant in a row and to nearby plants in the next row [6]. It is not excluded, however, that incidentally, progeny tubers may have become infected by inoculum migrating from infected haulms.

In 2020, at grower C, a relatively high percentage of the (composite) tuber samples (21%) and an estimated 5% of the tubers were infected with *D. fangzhongdai*. The presence was evidenced after the isolation of the pathogen by *gapA* sequencing and the use of a target-specific TaqMan assay. The pathogen was not detected in leaves or stems. *D. fangzhongdai* was recorded for the first time in 1973 in China, where it caused a bleeding canker disease in pear [34], and since then, in various ornamentals and woody plants in Asia and Europe, where it caused soft rot [2]. It has also been isolated from surface water in Scotland [35] but never found in association with potatoes. As for grower D in 2019, it may be that infections originated from the debris of (ornamental) plants in the soil. In the previous growing season, wheat was grown, while during wintertime, a green manure crop was grown and plowed under (personal communication grower).

Using a two-step analysis, first, a multiplex assay detecting all blackleg-causing species, followed by a characterization of the causative agent using simplex TaqMan assays for individual species, worked efficiently. Most infections found in the multiplex assay could be traced back.

The chance to isolate BL-SRP from latently infected plant samples positive in the enrichment TaqMan assays seems to depend on the enrichment procedure. We were more successful using extracts from vacuumed plant tissues after incubation than using incubation in a pectate enrichment broth under low oxygen conditions. The efficiency may also be dependent on the pathogen present. In 2019, attempts were made to isolate from samples in which *D. zeae* and *D. chrysanthemi* were detected, whereas, in 2020, samples were used in which *P. brasiliense* and *D. fangzhongdai* were detected by the enrichment TaqMan assay.

At this moment, only incidentally new SRP variants are found in tubers such as *D. zeae*, *D. chrysanthemi*, and *D. fangzhongdai*, but still, they may pose a threat to seed potato production. In this study, for *D. fangzhongdai*, it was shown that under experimental conditions, it is highly virulent and can cause blackleg at a similar level as *P. brasiliense*, but up to now, it has never been isolated from diseased plants in farmers’ fields in Europe. The existence of blackleg-causing species within the genus *Pectobacterium*, other than *P. atrosepticum*, *P. brasiliense*, and *P. parmentieri,* cannot be excluded. However, in surveys in the Netherlands on blackleg-diseased plants, these three *Pectobacterium* species were dominant as causative agents [3].

The occurrence of new pathogenic variants is likely. In a recent analysis of the pangenome of SRP, it was demonstrated that the evolution of soft rot Pectobacteriaceae is a highly dynamic process, which includes gene acquisitions partly in clusters, genome rearrangements, and loss of genes [23]. Variants may develop with the ability to cause blackleg, as has been found for *D. solani* and *P. brasiliense* [23,24,36]. This may result in the need for improved diagnostic tools, including the development of new TaqMan assays, to cover the broad spectrum of all blackleg-causing variants.

In conclusion, we found that before haulms are destructed, already high infection incidences can occur in a potato crop grown from minitubers with SRP, although the risks seem to be dependent on the environmental conditions during the growing season. Data suggests that the transmission pathway of the pathogen for haulm infection is different from that of tubers. The detection of various *Dickeya* species in crops indicate the risks for the occurrence of new SRP variants in potato and emphasizes the need for regular surveys.

## Figures and Tables

**Figure 1 microorganisms-10-02504-f001:**
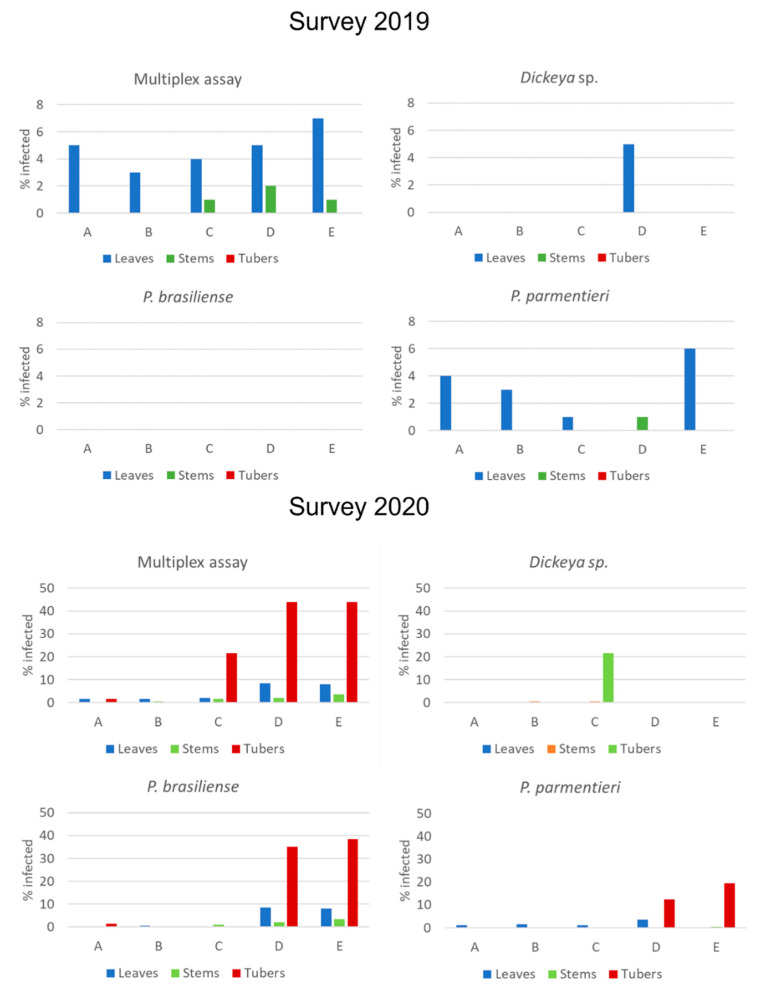
Results of an analysis of leaves, stems, and tubers of potato plants grown from minitubers at five growers (A–E) with a multiplex TaqMan assay detecting simultaneously *Dickeya* sp., Pectobacterium brasiliense, P. parmentieri, and P. atrosepticum followed by identification of positive results using simplex assays against the target pathogen. In 2019, 100 individual plants per grower were sampled. In 2020, 200 plants were sampled per grower.

**Figure 2 microorganisms-10-02504-f002:**
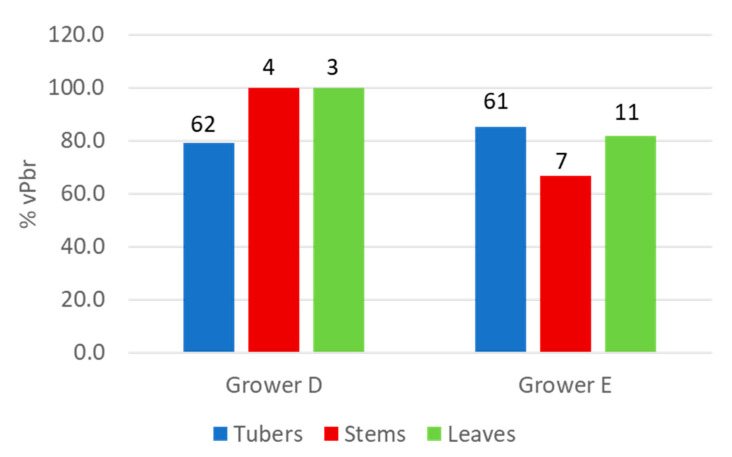
Percentage of samples (tubers, stems, or leaves) of a potato crop grown from minitubers positive in a TaqMan assay for a homogeneous clade of predominantly highly virulent *P. brasiliense* (vPbr TaqMan) calculated of the total number of positive samples in a Pbr TaqMan assay detecting all *P. brasiliense* strains. Number of samples positive with the Pbr TaqMan is indicated on top of the bars.

**Figure 3 microorganisms-10-02504-f003:**
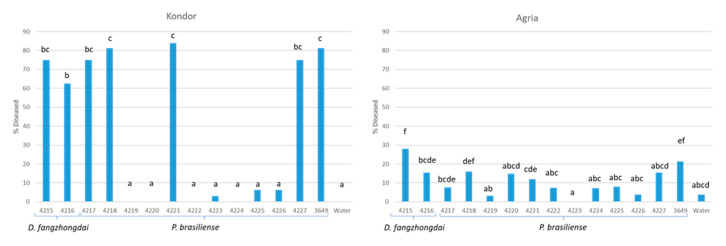
Disease prevalence of a potato crop after vacuum-infiltration of seed tubers of cv. Kondor and cv. Agria with *Dickeya fangzhongdai* or *Pectobacterium brasiliense* strains. Bars labelled with an identical character are not significantly different (*p* = 0.05).

**Figure 4 microorganisms-10-02504-f004:**
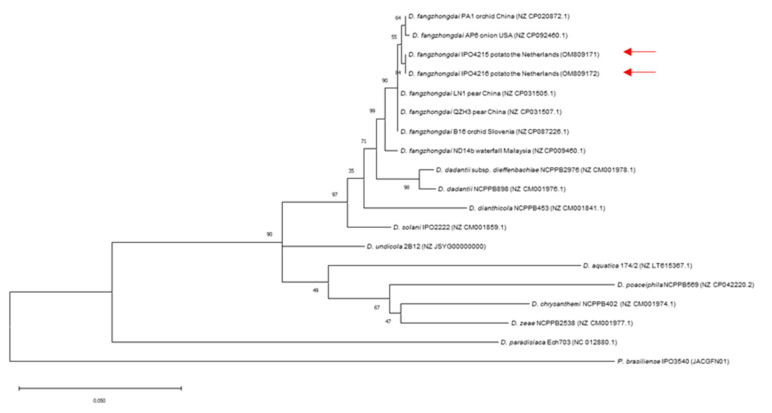
Maximum-likelihood phylogenetic tree of gapA sequences (762 bp) of *Dickeya* strains for identification of *D. fangzhongdai* (indicated with a red arrow), isolated from potato plants in the Netherlands. Bootstrap values greater than 40% are shown for 500 replicates. Concatenated sequences of *Pectobacterium brasiliense* were used as an out-group.

**Table 1 microorganisms-10-02504-t001:** Data on sampled growers of minitubers and sampling dates.

Grower	Region	Sampling Date	
		2019	2020
A	Noordoostpolder	25 June	17 June
B	Noordoostpolder	27 June	25 June
C	Noord-Holland	4 July	2 July
D	Noord-Holland	9 July	15 July
E	Noord-Holland	17 July	16 July

**Table 4 microorganisms-10-02504-t004:** Results of leaf samples collected at grower D in 2019 were positive with a genus-specific *Dickeya* TaqMan assay and tested with six *Dickeya* species-specific TaqMan assays.

	TaqMan Assay (Ct-Values)						
Sample	*Dickeya* sp.	*D. solani*	*D. dianthicola*	*D. dadantii*	*D. dieffenbachiae*	*D. chrysanthemi*	*D. zeae*
979	30	-	-	-	-	30.4	-
1018	30.4	-	-	-	-	-	31.3
1081	24	-	-	-	-	24.5	-
1141	23.9	-	-	-	-	24.4	-
1198	20.5	-	-	-	-	-	21.4

- = no reaction after 40 cycles.

## Data Availability

The data that support the findings of this study are available from the corresponding author upon reasonable request.

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
