# Peer review of "Natural Infections of Potato Plants Grown from Minitubers with Blackleg-Causing Soft Rot Pectobacteriaceae"

_microorganisms, 2022, doi:10.3390/microorganisms10122504_

Round 1
Reviewer 1 Report
Major concern:
1. From the abstract and introduction, the aim of the manuscript was tried to study the source of initial infection and the transmission pathway of potato blackleg causing SRP, however, the experimental design and results did not match these two issues.
2. The important part of the manuscript was to explore the source of initial infection, the different TaqMan assays were conducted. Since the probes and primers used in the manuscript were based on the reported species of SRP, there is no chance to find new species which might the initial infection.
3. Based on the assumption of the transmission pathway of the pathogen, from the potato haulms to the tubers or from leaf to soil, it is highly recommended to use labeled strain to carry out the experiment.
Minor concern:
Line 8-20, the Abstract Part is just a pile of results, there is no enough conclusive content.
Line 52-53, “five growers” should be explained in more detail when “grower” appeared in the first time, it seems that they are all experimental sites, not normal farmer lands.
Lines 64,106-107, 118, 180, there are four different temperatures ranging from 21°C to 28°C were used to culture the bacteria, “bacteria were grown on TSA for 48–72 h at 27 °C”, “Vacuum was applied to the bags and incubated flat at 25°C in a box in the climate room for 5 days”, “a sample of the leaf, stem or tuber extract was transferred from the vacuum bag to a tube with 900 μL PEB with screw cap and incubated for 1 day at 21°C”, “plates were incubated for 4 days at 28 °C”. Why?
Line 68, are the potato cultivars Agria and Kondor resistant or susceptible to SRP pathogens?
Line 70, “two crops” should be “two growers”.
Line 71, “three” and “the other three growers”.
Line 74, the sentence “200 plants per grower” seems not finished.
Line 101, “K2HPO4.3H2O” should be “K2HPO4·3H2O”.
Line 126, what is the function of adding Acidovorax cattleya before DNA-extraction? Should add more explaination.
Line 173, in the Table 2, the target of primers and probes in Dickeya sp. was not “potassium transporter Kup”, it should be “the up non-coding sequence of potassium transporter Kup”. The target of primers and probes in P. parmentieri did not perfectly distinguish P. wasabiae from itself by NCBI BLAST, so P. wasabiae in the environment would interfere the accurate result.
Line 194, “assay as described by12” should be “assay as described by Czajkowski12”. Same as line 197.
Line 197, “The ability”, whose ability?
Line 120, in 2019, “no target bacteria detected in the TaqMan assays could be isolated”, so was the data of 2019 accurate to support your conclusion?
Lines 237-238, primers and probes in P. parmentieri did not distinguish P. wasabiae from itself, other methods such as gapA phylogenetic tree should be re-conducted to confirm that the strains were parmentieri and no P. wasabiae disrupted the results.
Line 249, The sentence “In total 400 tubers were tested in 40 subsamples of 10 tubers” is hardly understood.
Line 262, “P. brasiliense” should be italics.
Line 262-263, the sentence “Tuber samples at these two growers plants existed on average of 7.4 to 6.2 tubers, respectively” is difficult to understand.
Line 265, what is the meaning of “infection incidence per tuber”?
Line 279, the sentence “P. atrosepticum was never detected” should be omitted.
Line 314, the isolated strains IPO4215 and IPO4216 were used in Figure 4 to construct maximum-likelihood phylogenetic tree but strains IPO4215 and IPO4222 in Figure S4 were used to macerate potato tuber tissue, why didn’t you use the same strains?
Line 316, “disease prevalence in cv. Kondor was much higher than in cv. Agria”, Figure 3 showed that this result was not only suitable for P. brasiliense, but also suitable for Dickeya fangzhongdai.
Line 320, in Figure 3, which strains were Dickeya fangzhongdai and which strains were Pectobacterium brasiliense? please indicate them.
Line 316, why there were no error bars in Figure 3?
Line 324, the strain names in the Figure 4 should be italic and the accession numbers of all strains used in phylogenetic tree should be indicated in the Figure.
Lines 329-358, the disadvantage of extracting DNA in 2019 and the advantage of the modified procedure of extracting DNA in 2020 (Lines 120-121) should be discussed.
Line 387, “This dissemination pathway has been evidenced before.”, please cite the reference.
Lines 403-406, please add more information in the Discussion Part, such as the fast changing of taxonomy and enough genomics will result in the inaccuracy of detecting species by TaqMan assays.
Lines 432-439, the Latin names should be italic.
Author Response
Dear reviewer,
Please, find our response in the attached file.
With kind regards,
Jan van der Wolf

Reviewer 2 Report
The article presents the results of a two-year monitoring of the spread of pathogens that cause soft rot in potatoes. The article has a lot of practical applications.
I have a few technical remarks.
1. What differences are taken into account significant at p=0.05 or p=0.01?
2. It is required to display the standard deviations and noteworthy differences in Figures 1, 2, 3, and S3.
3. The Figure 3 is referred to as figure 6 in the archive.
4. Figure 2 is not understandable. This information should be presented in a clearer manner.
5. Figures are not put in the text according to their initial citation sequence. Information is hard to comprehend when you have to seek for the appropriate graph for a long time.
6. Figure 1S's title indicates to Dickeya sp., and the image itself indicates Dickeya solani.
7. Probably, index 4 in table S3 refers to the ND value in row 11.
8. The colors used in Figure 1 in Survey 2020 for Dickeya sp. to represent leaves, stems, and tubers are different from those used in the other graphs.
9. You should check the links. The seventh link is lacking the journal's title.
Author Response
Dear reviewer,
Please, find our response in the file attached.
With kind regards,
Jan van der Wolf

Reviewer 3 Report
This manuscript reports the results of molecular diagnostic tests of potato plants that were growing at five farms in the Netherlands. One hundred plants from each farm were chosen and three plant tissues (leaves, stems, and tubers) were tested. The potato plants sampled in this study were derived from presumably healthy mini tubers. This study was conducted to ascertain the primary route that soft rot bacteria are introduced into potato cropping systems.
This study was a survey that was meant to determine if soft rot pathogens are introduced through the areal parts of the plant or through the soil. The bacterial species associated with each tissue was also determined, as it is likely that the there are species specific modes of transmission, tissue tropism, and disease symptoms. All of this information has value for determining how to most effectively allocate resources to target control strategies for the diseases caused by these bacteria. Additionally, it was informative to learn that Dickeya fangzhongdai is present in the Netherlands.
General criticisms
The source(s) of the mini tubers was not discussed, nor were there evaluations of the soft rot pathogen presence in the mini tubers that were sown for the plants in this study. Instead, the presence of soft rot pathogens in the tubers, mother tuber, and possibly stems must be considered to be the sum of that present in the seed tubers and that acquired after planting at the farms. This point should be made clear so to understand the limitations on the interpretations. It would also be helpful to know the disease incidence ratings as determined by visual inspection of fields? Was there a correlation between visual symptoms and molecular results?
It is not possible to appreciate the variance due to sampling or detection. A strategy to incorporate biological or technical replicates would allow the reader to assess how accurately the sampling represents the actual bacterial loads in the field and how well the DNA extraction and PCRs detect each of the species. The strategy used for Dickeya spp., which used a genus-specific test that was followed by species-specific tests, should be applied for Pectobacterium. It is not clear whether additional Pectobacterium species were present but not detected by the set of primers used.
Minor points
Line 50-56 - Please add rationale for analyzing stems.
Line 209 – Please add description of variables N and p.
Line 147 – What does the sample consist of? Is this purified bacterial DNA? Total DNA from plants?
Line 219 – Please define Rn values. Is the greater sensitivity of the FAM labeled primers a function of the label or the primer sequence?
Line 285 – Please add reference for “virulent clade”
Author Response
Dear reviewer,
Please find our response on your comments in the file attached.
With kind regards,
Jan van der Wolf

Round 2
Reviewer 3 Report
My concerns were addressed. Please review for consistency, for example, the change to BL-SRP.